# ParamMem: Augmenting Language Agents with Parametric Reflective Memory

**Tianjun Yao** [1 2]  **Yongqiang Chen** [2 3]  **Yujia Zheng** [3]  **Pan Li** [4]  **Zhiqiang Shen** [2]  **Kun Zhang** [2 3]

## Abstract

Self-reflection enables language agents to iteratively refine solutions, yet often produces repetitive outputs that limit reasoning performance. Recent studies have attempted to address this limitation through various approaches, among which increasing reflective diversity has shown promise. Our empirical analysis reveals a strong positive correlation between reflective diversity and task success, further motivating the need for diverse reflection signals. We introduce `ParamMem`, a parametric memory module that encodes cross-sample reflection patterns into model parameters, enabling diverse reflection generation through temperature-controlled sampling. Building on this module, we propose `ParamAgent`, a reflection-based agent framework that integrates parametric memory with episodic and cross-sample memory. Extensive experiments on code generation, mathematical reasoning, and multi-hop question answering demonstrate consistent improvements over state-of-the-art baselines. Further analysis reveals that `ParamMem` is sample-efficient, enables weak-to-strong transfer across model scales, and supports self-improvement without reliance on stronger external model, highlighting the potential of `ParamMem` as a effective component for enhancing language agents.[1]

## 1. Introduction

Large language models (LLMs) (Brown et al., 2020; Chowdhery et al., 2023; Touvron et al., 2023) have exhibited remarkable progress in complex reasoning tasks. A key insight driving recent advances is test-time scaling, i.e., allocating additional computation during inference to improve

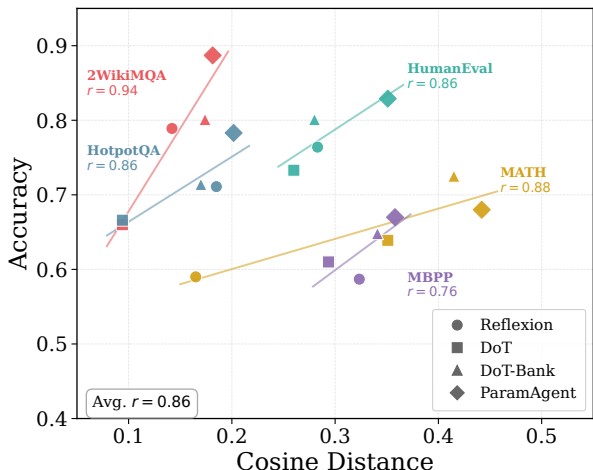

*Figure 1.* Correlation between reflective diversity (measured by average pairwise cosine distance) and task performance across five datasets using LLaMA-3.1-8B under Reflexion, DoT, and DoT-bank.

reasoning (Wei et al., 2022; Wang et al., 2022; Madaan et al., 2023; Yao et al., 2023a; Shinn et al., 2023; Snell et al., 2024). Among these approaches, reflection-based frameworks have proven particularly effective, where agents verbally reflect on task feedback and accumulate self-reflections in episodic memory to guide subsequent trials (Shinn et al., 2023; Madaan et al., 2023; Yao et al., 2023a). Such reflection mechanisms have been successfully applied to programming (Shinn et al., 2023), mathematical reasoning (Lightman et al., 2023), decision-making (Yao et al., 2023b), and multi-agent systems (Wu et al., 2023; Hong et al., 2024).

However, recent studies have identified limitations in self-reflection, showing that it often produces repetitive and inaccurate outputs (Huang et al., 2023; Yao et al., 2023c; Lingam et al., 2025; Ozer et al., 2025), which hinders the effectiveness of self-reflection. Among these works, Lingam et al. (2025) attempts to increase reflective diversity through prompt-level modifications (DoT) and by incorporating cross-sample trajectories (DoT-bank), demonstrating preliminary success. In this work, we first explore how reflective diversity relates to final performance. Specifically, we conduct experiments on five datasets using LLaMA-3.1-8B, computing the pairwise cosine distance across multi-round reflection logs for each sample under Reflexion, DoT, and DoT-bank, and averaging these distances. As illustrated in

---

[1]Shenzhen Loop Area Institute [2]Mohamed bin Zayed University of Artificial Intelligence [3]Carnegie Mellon University [4]Georgia Institute of Technology.

*Proceedings of the $43^{rd}$ International Conference on Machine Learning*, Seoul, South Korea. PMLR 306, 2026. Copyright 2026 by the author(s).

[1]Code can be found at: `https://github.com/tianyao-aka/ParamAgent`

Figure 1, the average Pearson correlation coefficient across the five datasets is 0.86, indicating a strong positive relationship between reflective diversity and task performance. While prompt-based approaches to diversifying reflections sometimes yield limited improvements, incorporating reasoning trajectories from similar samples often enhances diversity and the final performance.

Despite its effectiveness, the retrieval-based approach like DoT-bank relies on embedding similarity to retrieve cross-sample trajectories, which has limited capacity for capturing compositional patterns (Nguyen & Yates, 2023; Weller et al., 2025); moreover, learned embeddings are prone to collapse into low-rank subspaces, reducing retrieval diversity (Guo et al., 2023). This naturally raises our question:

> How can we further expand reflective diversity to achieve stronger reasoning performance?

To address this challenge, we introduce 🌐**ParamMem**, a new form of reflective memory that provides diversity through a fundamentally different mechanism. Unlike approaches that rely on prompt variations and retrieval-based methods that explicitly utilize similar samples, ParamMem operates by fine-tuning a lightweight parametric module on an auxiliary reflection dataset $\mathcal{D} = \{(x_i, r_i^g)\}_{i=1}^n$. Through training, the module encodes cross-sample patterns into its parameters; at inference time, it generates reflections by generalizing from these learned patterns rather than retrieving existing examples.

**Contribution.** We propose a new paradigm for enhancing reflective diversity to improve reasoning in language agents. Central to our approach is ParamMem, a parametric memory module that internalizes cross-sample reflection patterns. ParamMem targets diversity, is lightweight, and can seamlessly integrate into existing reflection-based frameworks. Building upon ParamMem, we propose ParamAgent and its enhanced variant ParamAgent-plus, which organically unify parametric reflective memory with episodic and cross-sample memory within a coherent framework. Through extensive empirical evaluation, our method exhibits several notable advantages: ① **Substantial performance gains.** Our approach achieves consistent improvements across various domains, outperforming state-of-the-art baselines significantly. ② **Sample efficiency.** ParamMem requires only ∼500 training samples to deliver strong performance, highlighting its effectiveness in low-data regimes. ③ **Self-improvement.** Even without relying on stronger external models, ParamMem can enhance reflective diversity using data generated by the base LLM itself, leading to improved performance for ParamAgent and ParamAgent-plus. ④ **Weak-to-strong transfer.** Even when ParamMem is trained using a weaker LLM, its generated reflective signals still enhance ParamAgent

built on stronger LLMs.

## 2. Preliminaries

We consider a pretrained language model $p_\theta$ that generates output $y$ given input $x$. We use $r_1, \ldots, r_k$ to denote self-reflections accumulated up to $k$ iterations, and use $r_k^g$ to denote the model-based outputs (e.g., reflections in programming and math tasks) sampled from the parametric memory module $\mathcal{M}_g$.

**Reflexion Framework.** Reflexion (Shinn et al., 2023) enables iterative reasoning through four components: (1) an **actor** $p_\theta$ that generates candidate solutions, (2) an **evaluator** that provides task-specific feedback (e.g., test results, correctness signals), (3) a **self-reflection module** $p_\theta$ that converts feedback into natural language reflections diagnosing errors, and (4) an **episodic memory** $\mathcal{M}$ that stores reflections from prior iterations. At iteration $k$, the actor generates candidate solutions conditioned on accumulated reflections from episodic memory:

$$y_k \sim p_\theta(\cdot \mid x, r_{1:k-1}). \tag{1}$$

**Cross-Sample Memory.** Cross-sample memory, which leverages past experiences or external logs, has been proposed in recent studies to enhance agent reasoning capabilities (Borgeaud et al., 2022; Shi et al., 2023; Wang et al., 2023; Zhong et al., 2024; Wang et al., 2024b). As recent studies have identified limited diversity in self-reflections, cross-sample memory is adopted to store reasoning trajectories from previously solved problems, thereby enriching the diversity of reflective inputs, which has proven effective in improving agentic reasoning. Given a new task, relevant trajectories are retrieved from the memory bank and incorporated into the prompt:

$$y \sim p_\theta(\cdot \mid x, r_{1:k}, \text{RETRIEVE}(\mathcal{B}, x)), \tag{2}$$

where $\mathcal{B}$ denotes the trajectory bank. In this study, we propose 🌐ParamMem, a parametric memory module $p_\phi(\cdot)$ complementary to episodic memory and cross-sample memory, which further promotes the diversity of reflective inputs. Based on ParamMem, we propose ParamAgent and ParamAgent-plus. In ParamAgent, the actor generates solutions conditioned on both episodic memory and parametric memory:

$$y_k \sim p_\theta(\cdot \mid x, r_{1:k-1}, r_k^g), \tag{3}$$

where $r_k^g \sim p_\phi(\cdot \mid x)$ denotes the reflection sampled from ParamMem at the $k$-th iteration. ParamAgent-plus further incorporates cross-sample memory, conditioning on all three memory sources:

$$y_k \sim p_\theta(\cdot \mid x, r_{1:k-1}, \text{RETRIEVE}(\mathcal{B}, x), r_k^g). \tag{4}$$

An architectural comparison of these frameworks is illustrated in Figure 2.

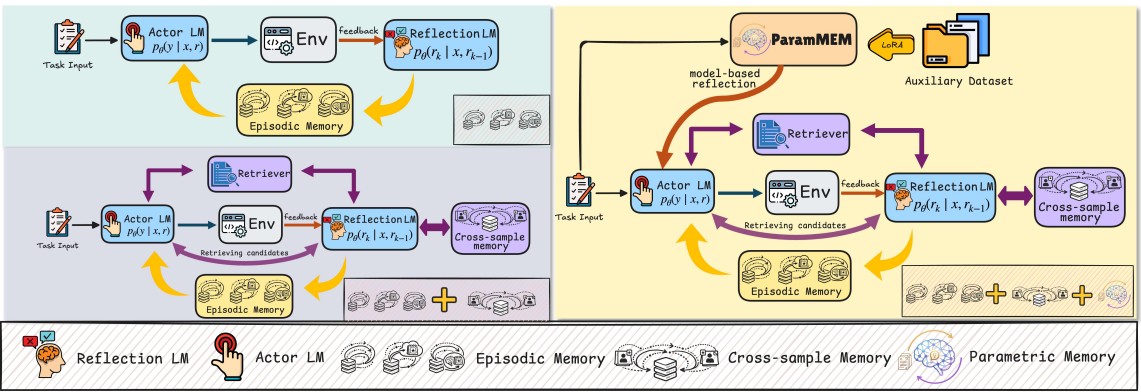

*Figure 2.* Comparison of memory mechanisms across different frameworks. Pale Mint denotes episodic memory only, as in Reflexion and DoT. Lavender Gray indicates episodic memory combined with cross-sample memory, as in DoT-bank. Soft Sand represents the full integration of episodic, cross-sample, and parametric memory, as in `ParamAgent-plus`. When using only episodic and parametric memory, the framework reduces to `ParamAgent`.

# 3. Augmenting Language Agents with 🧠**ParamMem**

In this section, we first describe how to construct `ParamMem`, and then present how to incorporate it into the proposed framework `ParamAgent`.

## 3.1. Building 🧠**ParamMem**

The core idea of `ParamMem` is to implicitly capture cross-sample regularities via training dynamics. Through fine-tuning, the module learns to generalize reflection patterns to unseen examples, rather than relying on prompt-based instructions or retrieving similar samples. While prompt-based methods are constrained by fixed instruction templates, and retrieval-based methods are limited by embedding similarity to existing examples, `ParamMem` can generate novel reflections by interpolating and extrapolating from learned patterns, therefore providing an additional source of diversity. The building process begins with constructing an auxiliary dataset for finetuning. Specifically, we curate a dataset $\mathcal{D} = \{(x_i, r_i^g)\}_{i=1}^n$, where $x_i$ denotes the input sample (e.g., a programming task), and $r_i^g = f_\phi(x_i; \mathcal{P})$ is obtained by prompting an LLM $f_\phi$ with a task-specific prompt $\mathcal{P}$ to generate auxiliary supervision for $x_i$. We then fine-tune a pretrained LLM on $\mathcal{D}$ using LoRA (Hu et al., 2022) to obtain the parametric module $\mathcal{M}_g$.

For **programming and math tasks**, $r_i^g$ takes the form of reflective feedback that enumerates potential mistakes and buggy implementations. For **multi-hop QA**, directly providing all supporting passages would consume excessive tokens. Inspired by cognitive chunking (Miller, 1956; Baddeley, 2020) and least-to-most prompting (Zhou et al., 2022), we instead prompt the LLM to decompose the query into compact semantic units and potential reasoning sub-tasks.

An example for programming and multi-hop QA is illustrated in Figure 3. Further details on dataset construction are provided in Appendix B.2.

---

**Algorithm 1** Pseudocode for the proposed method

---

**Require:** Dataset $\mathcal{D}$, base LM $p_\theta$, episodic memory $\mathcal{M}$, parametric module $\mathcal{M}_g$ parametrized by $p_\psi$, cross-sample memory bank $\mathcal{B}$, Failed task set $\mathcal{F}$, max iterations $T_{\max}$.
1: $\mathcal{M} \leftarrow \emptyset, \mathcal{B} \leftarrow \emptyset, F \leftarrow \emptyset$
    **Phase 1: `ParamAgent`**
2: **for** $x \in \mathcal{D}$ **do**
3:     **for** $t = 1$ to $T_{\max}$ **do**
4:         $T \leftarrow 0.2$ if $t = 1$, else $1.0$
5:         $r_t^g \sim p_\psi(\cdot \mid x; T)$ {Sample from $\mathcal{M}_g$}
6:         $r_{1:t-1} \leftarrow$ RETRIEVEREFLECTIONS$(\mathcal{M}, x)$
7:         $y_t \sim p_\theta(\cdot \mid x, r_{1:t-1}, r_t^g)$
8:         **if** EVALUATE$(y_t, x)$ **then**
9:             $\mathcal{B} \leftarrow \mathcal{B} \cup \{(x, \tau)\}$; **break** {Store trajectory}
10:       **else**
11:         $r_t \leftarrow$ GENERATESELFREFLECTION$(y_t)$
12:         $\mathcal{M} \leftarrow \mathcal{M} \cup \{(x, r_t)\}$
13:       **end if**
14:     **end for**
15:     **if** not solved **then** $\mathcal{F} \leftarrow \mathcal{F} \cup \{x\}$
16: **end for**
    **Phase 2: `ParamAgent-plus`** {Reattempt with cross-sample memory}
17: **for** $x \in \mathcal{F}$ **do**
18:     $\tau_{1:j} \leftarrow$ RETRIEVESIMILAR$(\mathcal{B}, x, j)$ {Retrieve $j$ trajectories}
19:     Repeat Phase 1 with $y_t \sim p_\theta(\cdot \mid x, r_{1:t-1}, r_{t-1}^g, \tau_{1:j})$
20: **end for**

---

## 3.2. Incorporating **ParamMem** into Reflexion-based Framework

Once the parametric module $\mathcal{M}_g$ is obtained, we incorporate it into the Reflexion-based framework. The integration is straightforward: at the $k$-th iteration, when providing

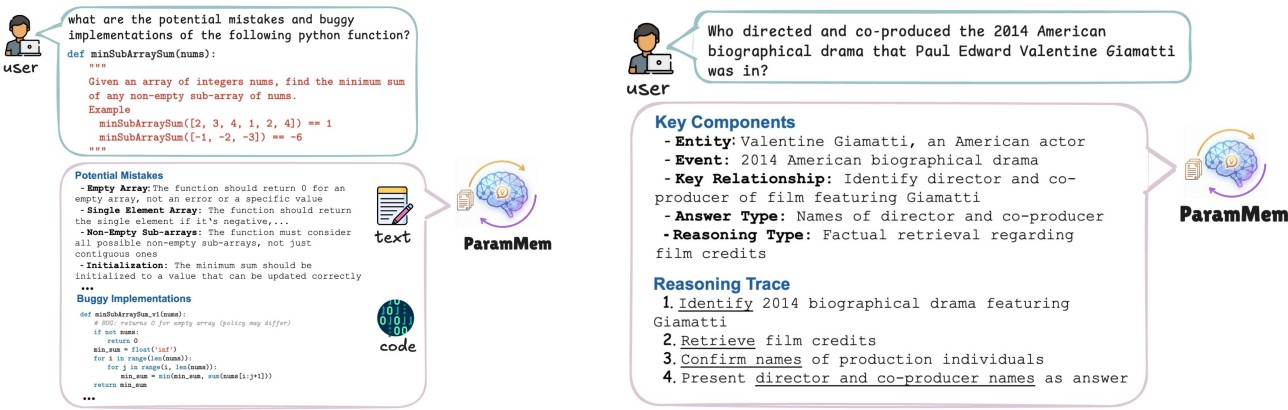

*(a)* An output example on programming task.

*(b)* An output example on multi-hop QA task.

*Figure 3.* Illustration of the output produced by 🌐ParamMem.

$\{r_1, \ldots, r_{k-1}\}$ to the actor, we additionally sample a model-based output $r_k^g \sim p_\psi(\cdot \mid x)$ from $\mathcal{M}_g$ and concatenate it with the self-reflections:

$$y_k \sim p_\theta(\cdot \mid x, r_{1:k-1}, r_k^g), \qquad (5)$$

where $r_k^g$ denotes the global-level reflection for programming and math, and denotes decomposed semantic unit and sub-tasks for multi-hop QA. We refer to this framework as ParamAgent. We further introduce ParamAgent-plus, a more powerful variant that additionally retrieves reasoning trajectories $\{\tau_1, \ldots, \tau_k\}$ from a memory bank $\mathcal{B}$ of previously solved tasks. To ensure a fair comparison, the retrieval mechanism follows DoT-bank (Lingam et al., 2025). The actor then conditions on both parametric and cross-sample signals:

$$y_k \sim p_\theta(\cdot \mid x, r_{1:k-1}, r_k^g, \tau_{1:j}). \qquad (6)$$

The pseudocode is provided in Algorithm 1. Similar to prior approaches (Yao et al., 2023c; Lingam et al., 2025), ParamMem does not directly interact with the environment during inference. Instead, by conditioning the actor on model-based feedback $r_k^g$, the output distribution is shaped by parametric knowledge, which subsequently influences the generation of new self-reflections. This feedback loop enables ParamMem to indirectly participate in the dynamic interaction process.

## 4. Experiments

In this section, we detail our experimental setup and present results across programming, math reasoning, and multi-hop QA. We then conduct more in-depth empirical analyses of our proposed method. Specifically, we first validate the effectiveness of ParamMem by analyzing how it promotes reflective diversity in both static and dynamic settings. We then perform comprehensive ablation studies to examine several key properties: (1) whether ParamAgent can achieve self-improvement without relying on stronger external models, (2) whether smaller parametric modules can enhance agents built on stronger LLMs (weak-to-strong transfer), and (3) the sample efficiency of ParamMem. More experimental results are included in Appendix B, including experiments with 70B scale LLMs and cost analysis.

### 4.1. Setup

**Datasets** We evaluate our framework across three domains. For programming, we use HumanEval (Chen et al., 2021) and MBPP (Austin et al., 2021). We also use Live-CodeBench (Jain et al., 2025), a more challenging dataset for additional empirical evaluation. For math reasoning, we adopt the MATH dataset (Hendrycks et al., 2021b), which covers competition-level problems of varying difficulty across seven subjects. For multi-hop QA, we use HotpotQA (Yang et al., 2018) and 2WikiMultiHopQA (Ho et al., 2020), which require reasoning across multiple passages. Further details about each dataset, as well as how we perform dataset splits are provided in Appendix B.

**Evaluation** For programming tasks, we report Pass@1. During generation, only visible or synthetic test cases are used, while final evaluation is conducted on hidden test cases; a score of 1 is assigned if all tests pass and 0 otherwise. For math reasoning and multi-hop QA, we report 0–1 accuracy on subsampled testsets.

**Baselines** We compare against: (1) **Base**, the underlying LLM agent without reflection; (2) **Reflexion** (Shinn et al., 2023), which uses episodic self-reflections; (3) **Retro-former** (Yao et al., 2023c), which also employs a parametric reflective module but trains it via policy gradient optimization to improve reflection accuracy rather than diversity, which serves as a direct comparison with ParamAgent; (4) **DoT** (Lingam et al., 2025), which augments Reflexion

with prompt-level diversity; (5) **DoT-bank** (Lingam et al., 2025), which further incorporates a memory bank to enrich the reflective feedbacks.

To ensure a comprehensive evaluation, we employ three backbone LLMs with varying levels of reasoning capability: (1) **Llama-3.1-8B** (Dubey et al., 2024), a strong open-source reasoning model; (2) **Mistral-7B-v0.2** (Jiang et al., 2023), a competitive medium-sized model with efficient inference; and (3) **Qwen2-1.5B-instruct** (Bai et al., 2023). This selection of backbones allows us to examine how our approach performs across different model sizes and reasoning strengths. We also provide results with stronger base LLMs in Appendix B.3, showing that even with a smaller backbone as the parametric module $\mathcal{M}_g$, it can still provide noticable gains to agents built on 70B-scale LLMs.

**Implementation details** Across all experiments, we fix the number of iterations to 5 for both baseline methods and our proposed approach. For `ParamAgent` and its variants, we set the sampling temperature to $T = 0.2$ during the first iteration, and $T = 1.0$ in the subsequent iterations to promote diversity. Unless otherwise specified, the parametric model is instantiated using Llama3.1-8B-Instruct, and is finetuned via LoRA. For LoRA finetuning, we use a rank of $r = 128$, scaling factor $\alpha = 32$, a learning rate of $2e - 5$, and train for 3 epochs.

### 4.2. Experimental Results

In this section, we first present the main results across 3 domains (Observation ①). We then analyze how `ParamMem` promotes reflective diversity in both static and dynamic settings (Observations ②), followed by a case study explaining why increased reflective diversity leads to performance gains (Observation ③). We then conduct ablation studies examining our framework without relying on stronger external models to generate datasets (Observation ④), iterative self-teaching (Observation ⑤), weak-to-strong transfer where smaller parametric modules enhance stronger agents (Observation ⑥), and the advantage of sample efficiency (Observation ⑦).

**Observation** ①: **`ParamMem` consistently enhances Reflexion-based frameworks across all domains.** As shown in Tables 1, both `ParamAgent` and `ParamAgent-plus` achieve remarkable performance across the three domains. We note that `ParamAgent` differs from Reflexion and DoT solely through the incorporation of `ParamMem`, while `ParamAgent-plus` extends DoT-bank by augmenting its episodic and cross-sample memory with our parametric module.

On programming benchmarks, `ParamAgent` achieves significant improvements over all baselines even without the cross-sample memory component. Similarly, on

multi-hop QA, `ParamAgent` substantially outperforms most prior methods, highlighting the standalone effectiveness of `ParamMem`. For mathematical reasoning, while `ParamAgent` improves upon Reflexion and DoT, we observe that cross-sample trajectories play a more critical role. This aligns with the intuition that mathematical problem-solving benefits from exposure to analogous problems and solution patterns, akin to how humans learn to solve math problems. Nevertheless, `ParamAgent-plus` still outperforms DoT-bank by incorporating `ParamMem`, demonstrating the complementary value of model-based reflective memory. Notably, Retroformer excels on MATH, where reflection accuracy may matter more than diversity. However, it underperforms on programming and multi-hop QA despite also using parametric encoding. We attribute this to distribution shift, as the training data may not align well with test data, causing accuracy-focused optimization to overfit. In contrast, `ParamMem`'s objective is diversity-driven, implying that diversity-focused parametric memory generalizes better across distributions.

**Observation** ②: **`ParamMem` induces an additional layer of reflective diversity beyond episodic and cross-sample memory.** We hypothesize that the parametric module $\mathcal{M}_g$ introduces an additional source of diversity through the training dynamics. To verify this, we conduct the following analysis on programming tasks. We fine-tune Llama-3.1-8B as the parametric module using synthetic datasets generated by either GPT-4o-mini or Llama-3.1-8B itself. For each task in HumanEval, we sample 10 reflections at temperature $T = 1.0$, embed the outputs, and compute the mean value of pairwise cosine distances $D_{mean}$, and the distribution across all samples. As illustrated in Figure 4, the parametric module trained on GPT-4o-mini data yields the highest diversity. Notably, even when using the same LLM for both data generation and fine-tuning, the resulting diversity still exceeds that of the unfinetuned Llama-3.1-8B. This finding also explains why `ParamAgent` and `ParamAgent-plus` remain effective in self-improvement settings (Table 2). In contrast, Retroformer exhibits diversity even lower than the frozen Llama-3.1-8B, suggesting that its policy-gradient training drives the module to converge to a single correct reflection mode for each sample rather than exploring diverse hypotheses.

The above analysis characterizes diversity in a static setting. We further examine whether this diversity persists when `ParamMem` is incorporated into the Reflexion-based framework and interacts with the environment via $p_\theta(\cdot \mid x, r_{1:k-1}, r_k^g)$. Specifically, we maintain the complete reflection history for each sample on HumanEval and embed all reflections using OpenAI text-embedding-3-small model (OpenAI, 2024). We then perform $K$-means clustering (Lloyd, 1982) over all reflections and apply the elbow method (Tibshirani et al., 2001) to determine the

*Table 1.* Performance on HumanEval/MBPP, MATH, HotpotQA, and 2WikiMultiHopQA. **Bold** denotes the best result, and underline marks the second best. ↑ and ↓ indicate the absolute improvement or decrease relative to the Base method. For clarity, the prompt token usage of the Base method is normalized to 1. *Score* is Pass@1 for HumanEval/MBPP and Accuracy for MATH/QA.

| Domain | Dataset | Method | Llama-3.1-8B | | Mistral-7B-v0.2 | | Qwen2-1.5B | |
|---|---|---|---|---|---|---|---|---|
| | | | Score | #Prompt Tokens | Score | #Prompt Tokens | Score | #Prompt Tokens |
| **Code** | **HumanEval** | Base | 59.15 | 1.00 | 32.93 | 1.00 | 41.46 | 1.00 |
| | | Reflexion | 76.22 ↑17.07 | 9.29 | 51.22 ↑18.29 | 28.54 | 49.39 ↑7.93 | 18.30 |
| | | Retroformer | 67.68 ↑8.53 | 11.28 | 42.94 ↑10.01 | 38.37 | 46.34 ↑4.88 | 12.77 |
| | | DoT | 73.17 ↑14.02 | 17.45 | 46.95 ↑14.02 | 43.06 | 56.56 ↑15.10 | 15.26 |
| | | DoT-bank | 79.56 ↑20.41 | 24.71 | 54.26 ↑21.33 | 61.62 | 60.10 ↑18.64 | 31.28 |
| | | ParamAgent | **82.93** ↑23.78 | 19.18 | **67.07** ↑34.14 | 70.38 | **66.46** ↑25.00 | 33.45 |
| | **MBPP** | Base | 47.61 | 1.00 | 24.94 | 1.00 | 42.06 | 1.00 |
| | | Reflexion | 58.69 ↑11.08 | 37.18 | 28.46 ↑3.52 | 14.02 | 47.61 ↑5.55 | 26.95 |
| | | Retroformer | 42.82 ↓4.79 | 8.64 | 21.66 ↓3.28 | 12.08 | 31.49 ↓10.57 | 23.70 |
| | | DoT | 61.21 ↑13.60 | 51.83 | 19.79 ↓5.15 | 25.45 | 47.37 ↑5.31 | 21.48 |
| | | DoT-bank | 64.82 ↑17.21 | 69.41 | 24.68 ↓0.26 | 60.09 | 53.38 ↑11.32 | 60.95 |
| | | ParamAgent | **67.00** ↑19.39 | 86.39 | **51.64** ↑26.70 | 36.88 | **54.90** ↑12.84 | 66.86 |
| **Math** | **MATH** | Base | 48.20 | 1.00 | 12.23 | 1.00 | 8.99 | 1.00 |
| | | Reflexion | 58.99 ↑10.79 | 23.33 | 19.78 ↑7.55 | 27.67 | 21.94 ↑12.95 | 18.39 |
| | | Retroformer | 63.67 ↑15.47 | 17.09 | **43.53** ↑31.30 | 35.67 | **33.09** ↑24.10 | 30.12 |
| | | DoT | 64.38 ↑16.18 | 34.17 | 23.25 ↑11.02 | 40.51 | 22.30 ↑13.31 | 31.99 |
| | | DoT-bank | 73.02 ↑24.82 | 83.92 | 35.61 ↑23.38 | 122.92 | 24.37 ↑15.38 | 76.71 |
| | | ParamAgent | 67.99 ↑19.79 | 57.01 | 28.06 ↑15.83 | 92.91 | 22.30 ↑13.31 | 70.07 |
| | | ParamAgent-plus | **75.45** ↑27.25 | 111.32 | 38.96 ↑26.73 | 196.18 | 25.97 ↑16.98 | 144.25 |
| **QA** | **HotpotQA** | Base | 57.67 | 1.00 | 45.00 | 1.00 | 43.66 | 1.00 |
| | | Reflexion | 71.33 ↑13.66 | 4.13 | 62.33 ↑17.33 | 4.67 | 50.03 ↑6.37 | 6.22 |
| | | Retroformer | 73.00 ↑15.33 | 2.77 | 67.33 ↑22.33 | 4.59 | 47.70 ↑4.04 | 9.17 |
| | | DoT | 66.67 ↑9.00 | 7.10 | 58.33 ↑13.33 | 8.97 | 49.32 ↑5.66 | 58.05 |
| | | DoT-bank | 72.00 ↑14.33 | 13.28 | 66.33 ↑21.33 | 19.35 | 52.02 ↑8.36 | 109.54 |
| | | ParamAgent | **78.33** ↑20.66 | 22.25 | **69.67** ↑24.67 | 34.99 | **64.66** ↑21.00 | 14.69 |
| | **2WikiMultiHopQA** | Base | 40.33 | 1.00 | 21.00 | 1.00 | 40.33 | 1.00 |
| | | Reflexion | 78.67 ↑38.34 | 5.47 | 61.33 ↑40.33 | 5.86 | 51.00 ↑10.67 | 6.56 |
| | | Retroformer | 77.00 ↑36.67 | 5.90 | 71.00 ↑50.00 | 5.33 | 67.66 ↑27.33 | 3.68 |
| | | DoT | 66.67 ↑26.34 | 7.03 | 52.13 ↑31.13 | 6.40 | 47.83 ↑7.50 | 30.55 |
| | | DoT-bank | 80.33 ↑40.00 | 12.49 | 74.66 ↑53.66 | 8.10 | 50.49 ↑10.16 | 54.92 |
| | | ParamAgent | **88.67** ↑48.34 | 10.41 | **81.33** ↑60.33 | 14.43 | 63.33 ↑23.00 | 17.39 |

optimal number of clusters $K^*$. As shown in Figure 4, ParamAgent achieves $K^* = 39$, substantially larger than Reflexion, DoT, and DoT-bank, indicating that **ParamMem** introduces significantly richer and more varied reflective signals. Moreover, the silhouette scores of ParamAgent are consistently higher across all $K$, confirming superior clustering quality and semantic coherence of the generated reflections, as illustrated in Figure 4. In conclusion, these analyses demonstrate that **ParamMem introduces a complementary source of reflective diversity**, thereby enriching the feedback signals available to the agent throughout iterative refinement.

**Observation ③: Diverse reflections enlarge the hypothesis space for error diagnosis.** To understand the reason behind diversity-driven gains, we conduct a case study on MBPP, focusing on instances where ParamAgent succeeds but Reflexion and DoT fail (Figure 8 in Appendix B.5). We observe that self-reflections often fail to identify the core source of errors and mislead the agent away from correct implementations. While ParamAgent is not immune to such

*Table 2.* Self-improvement results using Llama-3.1-8B as both the agent and data generator **Bold** denotes the best, underline the second best. 🤖 denotes Llama-3.1-8B-Instruct as the data generator.

| Method | HumanEval | HotpotQA |
|---|---|---|
| Base | 59.15 | 57.67 |
| Reflexion | 76.22 ↑17.07 | 71.33 ↑13.66 |
| DoT | 73.17 ↑14.02 | 66.67 ↑9.00 |
| DoT-bank | 79.56 ↑20.41 | 72.00 ↑14.33 |
| ParamAgent 🤖 | 78.05 ↑18.90 | 76.33 ↑18.66 |
| ParamAgent-plus 🤖 | **86.59** ↑27.44 | **83.33** ↑25.66 |

failure modes, the increased diversity of reflective feedback provides the agent with a broader set of diagnostic hypotheses, thereby increasing the likelihood of encountering the correct cue for successful refinement. This also explains why ParamAgent and ParamAgent-plus occasionally incur higher token consumption in certain datasets.

**Observation ④: ParamMem supports agent self-improvement without dependence on stronger external**

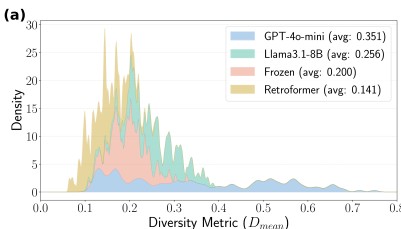 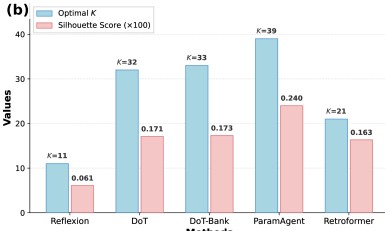 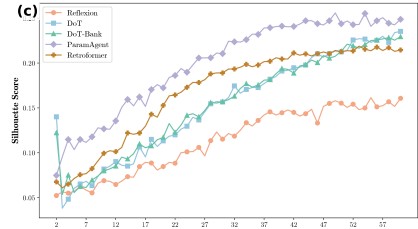

*Figure 4.* Reflection diversity induced by 🌐`ParamMem`. (a) Higher pairwise distance indicates more diverse outputs. (b) Higher optimal $K$ and silhouette scores confirm greater semantic variation. (c) Silhouette score as a function of cluster number $K$.

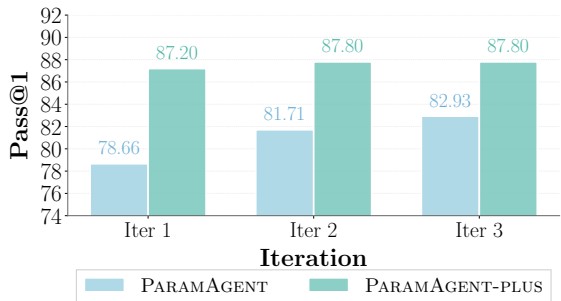

*Figure 5.* The performance of `ParamAgent` and `ParamAgent-plus` in HumanEval dataset, with 3 iterative process.

*Table 3.* Weak-to-strong transfer on LiveCodeBench, HumanEval, and HotpotQA with **Qwen3-Next-80B-A3B-Instruct** as the base LLM in the agent. 🧠 and 🦜 denote `ParamMem` instantiated by Llama-3.1-8B-Instruct and Qwen3-Next-30B-A3B-Instruct.

| Method | LiveCodeBench | HumanEval | HotpotQA |
|---|---|---|---|
| Simple | 52.00 | 90.24 | 71.33 |
| Reflexion | 62.00 ↑10.00 | 96.34 ↑6.10 | 82.33 ↑11.00 |
| DoT | 60.67 ↑8.67 | 96.34 ↑6.10 | 79.67 ↑8.34 |
| DoT-bank | 62.00 ↑10.00 | 96.95 ↑6.71 | 83.33 ↑12.00 |
| `ParamAgent` 🧠 | 61.33 ↑9.33 | 97.56 ↑7.32 | 79.67 ↑8.34 |
| `ParamAgent-plus` 🧠 | 63.33 ↑11.33 | 98.17 ↑7.93 | **85.00** ↑13.67 |
| `ParamAgent` 🦜 | 64.67 ↑12.67 | - | 76.33 ↑5.00 |
| `ParamAgent-plus` 🦜 | **68.00** ↑16.00 | - | 83.67 ↑12.34 |

**models.** Recent work on self-improving agents seeks to enhance reasoning capabilities without relying on external stronger models (Wei et al., 2022; Zelikman et al., 2022; Shinn et al., 2023; Zeng et al., 2024; Snell et al., 2025; Muennighoff et al., 2025). `ParamAgent` exhibits a similar property: even when `ParamMem` is fine-tuned on synthetic data generated by the base model itself, it still yields consistent gains. Specifically, we use Llama-3.1-8B to generate synthetic data and fine-tune the same base model as the parametric memory module. As shown in Table 2, `ParamAgent` and `ParamAgent-plus` improve significantly over DoT and DoT-bank, respectively. This demonstrates that `ParamMem` can enhance reasoning through diversified reflections without relying on stronger external models.

**Observation ⑤: Iterative self-teaching further enhances `ParamAgent`.** We investigate iterative self-teaching on HumanEval: starting from Llama-3.1-8B-Instruct, we fine-tune `ParamMem` on 1,000 randomly sampled examples. After training, we use the resulting model to generate new targets for the same inputs, yielding an updated dataset $\mathcal{D}' = \{(x_i, \tilde{r}_i^g)\}_{i=1}^{1000}$ where $\tilde{r}_i^g \sim p_\phi(\cdot \mid x_i)$ are freshly sampled reflections. We then fine-tune on $\mathcal{D}'$ and repeat this process for 3 iterations. As shown in Figure 5, `ParamAgent` improves steadily across iterations, suggesting that `ParamMem` progressively produces more diverse reflections. In contrast, `ParamAgent-plus` shows only

marginal gains, we hypothesize that with cross-sample trajectories, the model already approach the diversity ceiling.

This resembles STaR (Zelikman et al., 2022) and recent variants (Hosseini et al., 2024; Zeng et al., 2024), but with a key difference: those methods filter for correct samples at each iteration, implicitly performing reward-driven bootstrapping. Our approach requires no filtering, since `ParamMem` targets diversity rather than correctness.

**Observation ⑥: `ParamAgent` enables weak-to-strong reasoning augmentation.** The previous observation demonstrates that `ParamMem` supports self-improvement. Here we examine whether a weaker model serving as `ParamMem` can enhance a stronger agent. We evaluate on Live-CodeBench and HotpotQA using Qwen3-Next-80B-A3B-Instruct (Team, 2025) as the base model in the agent, with `ParamMem` instantiated by either Llama-3.1-8B or Qwen3-Next-30B-A3B-Instruct (Team, 2025). As shown in Table 3, both configurations consistently outperform all baselines. For coding, the Qwen3-30B-based `ParamMem` proves more effective, improving over the strongest baseline by 9.7%. Interestingly, for multi-hop QA, the smaller Llama-3.1-8B-based `ParamMem` outperforms its Qwen3-30B counterpart, achieving a 2.0% relative improvement over the best baseline method. These results confirm that `ParamMem` enables weak-to-strong transfer: even smaller models can provide diverse reflective signals that benefit stronger agents.

**Observation ⑦: `ParamMem` is sample-efficient.** We

*Table 4.* Sample efficiency of 🌐 ParamMem. Models are trained on 500 diverse samples via $K$-means clustering. **Bold** denotes the best result, underline the second best.

| Method | HumanEval | MBPP |
|---|---|---|
| DoT | 73.17 | 61.21 |
| DoT-bank | 79.56 | 64.82 |
| ParamAgent ( 8000 samples) | 82.93 | **67.00** |
| ParamAgent (500 samples) | 81.71 | 64.99 |
| ParamAgent-plus (500 samples) | **86.59** | 65.49 |

examine how many samples are needed for an effective ParamMem. We apply $K$-means clustering to the GPT-4o-mini synthetic data and sample 500 diverse examples across clusters for fine-tuning. As shown in Table 4, ParamAgent and ParamAgent-plus retain strong performance on HumanEval and MBPP even with this reduced training set; Notably, ParamAgent-plus with only 500 training samples outperforms ParamAgent trained on the dataset of over 8000 samples, demonstrating the effective synergy of episodic, cross-sample, and parametric memory.

# 5. Related Work

**LLM Reasoning and Diversity** LLMs perform multi-step reasoning through techniques like CoT prompting (Wei et al., 2022), Self-Consistency (Wang et al., 2022), and Re-Act (Yao et al., 2023b). Self-Consistency improves CoT by sampling multiple reasoning paths and aggregating them via majority voting, demonstrating that diversity in reasoning traces leads to more robust outputs. Iterative self-feedback methods (Madaan et al., 2023; Shinn et al., 2023) and test-time compute scaling (Snell et al., 2025; OpenAI et al., 2024; Guo et al., 2025) further improve reasoning by allocating additional computation during inference. To enhance diversity, structured exploration methods like Tree of Thoughts (Yao et al., 2023a) and Graph of Thoughts (Besta et al., 2024) enable deliberate search over reasoning states. These methods highlight that exploring diverse solution paths is crucial for solving complex problems. Most relevant to our work, DoT (Lingam et al., 2025) addresses repetitive self-reflections via prompt-level interventions and cross-sample memory. We extend this line of research by proposing ParamMem, which provides an orthogonal source of reflective diversity beyond episodic and cross-sample memory through parametric encoding.

**Improving Reflection in LLM Agents** Recent work has explored diversity-driven approaches to improve reflection. DoT (Lingam et al., 2025) addresses repetitive self-reflections through prompt-level interventions and cross-sample memory retrieval. Beyond diversity-driven methods, other approaches improve reflection through different mechanisms. Retroformer (Yao et al., 2023c) uses policy gradi-

ent optimization to learn a retrospective model that refines prompts based on environment feedback, enabling the agent to improve its reflection accuracy over time. Self-RAG (Asai et al., 2024) trains special reflection tokens directly into the generative model, enabling self-critique of generation and retrieval decisions during inference. ExpeL (Zhao et al., 2024) extracts generalizable insights from successful trajectories and stores them for cross-task transfer. These approaches primarily focus on improving reflection quality or accuracy through various mechanisms. While ParamMem also adopts parametric encoding of reflections like Retroformer and Self-RAG, it differs in both purpose and design: rather than optimizing reflection accuracy or self-critique capability, ParamMem aims to enhance reflective diversity by unifying episodic memory, cross-sample memory, and parametric memory within a single framework.

**Self-improving Language Agents** Self-improvement in language models enables agents to enhance their reasoning capabilities through iterative learning from self-generated data. STaR introduced bootstrapped reasoning, where models generate rationales and fine-tune on those leading to correct answers (Zelikman et al., 2022). This paradigm was extended by ReST (Gulcehre et al., 2023) and ReST-EM (Singh et al., 2023), which demonstrated that self-generated training data can surpass human-annotated data when verifiable feedback is available. More recent work has eliminated the need for external reward models entirely: Self-Rewarding Language Models (Yuan et al., 2024) use LLM-as-a-Judge (Zheng et al., 2023) to generate preference data, while Meta-Rewarding (Wu et al., 2025) adds meta-judgment capabilities. In contrast to these approaches, ParamMem enables self-improvement by progressively diversifying reflective feedback, thereby strengthening reflection-based frameworks.

# 6. Conclusions and Limitations

We propose 🌐**ParamMem**, a parametric memory module that internalizes reflections into model parameters, inducing an additional layer of diversity beyond episodic and cross-sample memory. Building upon ParamMem, we introduce ParamAgent and ParamAgent-plus, which augment reflection-based reasoning frameworks with ParamMem. Across 3 domains, our methods deliver substantial performance gains over state-of-the-art baselines, highlighting the potential of parametric memory as a lightweight plug-in module for building language agents. Despite these advantages, our approach has limitations. A notable one is the increased token consumption in certain scenarios, which is an inherent cost of the additional reflective diversity. In future work, we aim to address this trade-off by exploring more token-efficient integration strategies.

## Impact Statement

Reflection-based reasoning is widely adopted in language agent systems to improve task performance across diverse domains. This work studies a fundamental limitation of self-reflection: the lack of diversity in generated reflections, and proposes a novel approach to address it. Our method has a positive impact on advancing the understanding and capability of language agents. We do not foresee any potential negative societal impact arising from this work.

## Acknowledgements

We would like to acknowledge the support from NSF Award No. 2229881, AI Institute for Societal Decision Making (AI-SDM), the National Institutes of Health (NIH) under Contract R01HL159805, and grants from Quris AI, Florin Court Capital, MBZUAI-WIS Joint Program, and the Al Deira Causal Education project.

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

# Appendix

## A. More Related Work

**Memory Systems for Language Agents.** Memory architectures have been extensively studied to enhance agent capabilities. Generative Agents (Park et al., 2023) introduced the memory stream with retrieval based on recency, importance, and relevance. MemGPT (Packer et al., 2023) applies OS-inspired virtual memory management, while MemoryBank (Zhong et al., 2024) incorporates Ebbinghaus forgetting curves for human-like memory decay. For cross-sample learning, ExpeL (Zhao et al., 2024) maintains experience pools and abstracted insight stores. Notably, nearly all existing memory systems are retrieval-based, relying on embedding similarity to access stored experiences. While retrieval-based approaches like DoT-bank (Lingam et al., 2025) have shown promise in diversifying reflections through cross-sample trajectories, they suffer from limited capacity for capturing compositional patterns (Nguyen & Yates, 2023) and embedding collapse into low-rank subspaces (Guo et al., 2023).

**Parametric Approaches to Learning from Experience.** A growing line of work explores encoding experiences and reflection capabilities directly into model parameters. Retroformer (Yao et al., 2023c) trains a dedicated retrospective model via policy gradient reinforcement learning to generate improved verbal feedback, demonstrating that learned reflection can outperform prompt-based approaches; however, it requires expensive online RL with environment interaction. Self-RAG (Asai et al., 2024) trains LLMs to generate special reflection tokens that enable adaptive retrieval decisions and self-critique of generation quality, representing a hybrid between prompting and parametric learning, though it focuses primarily on factual verification rather than reasoning diversity. LEMA (An et al., 2023) fine-tunes LLMs on mistake-correction pairs generated by stronger models, achieving strong results on mathematical reasoning by parametrically encoding error patterns. SCoRe (Kumar et al., 2024) trains models for intrinsic self-correction via multi-turn reinforcement learning, demonstrating significant improvements without external feedback but at considerable computational cost. MemoryLLM (Wang et al., 2024a) integrates a self-updatable memory pool within the LLM's latent space, enabling parametric storage without full fine-tuning. These works collectively suggest that parametric learning can be more effective than retrieval for capturing complex patterns, though often at significant computational cost or with narrow task focus. Our work introduces `ParamMem`, a lightweight parametric memory module that specifically targets reflective diversity, encoding cross-sample reflection patterns into parameters via efficient supervised fine-tuning to enable diverse reflection generation, which supports reflection-based framework to unifies various forms of memories.

## B. More Experimental Details Results

### B.1. Dataset Statistics

**Programming.** For programming tasks, we evaluate on HumanEval (Chen et al., 2021) and MBPP (Austin et al., 2021). HumanEval consists of 164 hand-written Python programming problems, each accompanied by hidden unit tests and a small number of visible test cases. We additionally consider MBPP, which provides 974 crowd-sourced Python problems; following prior work, we use the 397 problems from the filtered evaluation split.

**Math.** For mathematical reasoning, we adopt the MATH dataset (Hendrycks et al., 2021b), which contains competition-style math problems spanning seven subjects including Algebra, Geometry, Number Theory, Counting and Probability, and Precalculus. We randomly sample a balanced subset across categories for evaluation.

**Multi-hop QA.** For multi-hop question answering, we use HotpotQA (Yang et al., 2018) and 2WikiMultiHopQA (Ho et al., 2020). In HotpotQA, we stratify by difficulty level and randomly sample 100 examples from each category (easy, medium, hard), yielding a total of 300 evaluation samples. For 2WikiMultiHopQA, we stratify by question type and randomly sample 75 examples from each of four categories (bridge comparison, comparison, compositional, inference), again yielding 300 samples in total. These stratified subsets ensure balanced evaluation across different reasoning styles.

### B.2. Finetuning the Parametric Module

**Programming** For programming tasks, we curate a dataset by sampling 4000 coding problems from the APP dataset (Hendrycks et al., 2021a) at introductory level. In addition, we synthesize 4200 problems using GPT-4o-mini, covering a diverse range of programming domains. The code templates and prompt used for data generation are provided in

*Table 5.* Datasets used for Programming, Math, and Multi-hop QA tasks.

| Task Type | Dataset Name | Size | Metric |
|---|---|---|---|
| Programming | HumanEval | 164 problems, $\sim$3 visible test cases/problem | Pass@1 |
| Programming | MBPP | 397 sampled problems | Pass@1 |
| Math | MATH | 278 sampled problems across 7 subjects | 0-1 Acc |
| Multi-hop QA | HotpotQA | 300 sampled problems (100 per difficulty) | 0-1 Acc |
| Multi-hop QA | 2WikiMultiHopQA | 300 sampled problems (75 per type) | 0-1 Acc |

*Table 6.* Performance on HumanEval. **Bold** denotes the best result, and underline marks the second best. ↑ and ↓ indicate absolute change relative to the Base method. For clarity, the prompt token usage of the Base method is normalized to 1.

| Dataset | Method | Llama-3.1-70B-Instruct | | Qwen2.5-72B-Instruct | |
|---|---|---|---|---|---|
| | | **Pass@1** | **#Prompt Tokens** | **Pass@1** | **#Prompt Tokens** |
| | Base | 80.49 | 1.00 | 82.92 | 1.00 |
| | Model-based Reflection | 87.80 ↑7.31 | 6.39 | 89.64 ↑6.72 | 3.48 |
| | Reflexion | 90.24 ↑9.75 | 4.31 | 88.41 ↑5.49 | 3.48 |
| **HumanEval** | DoT | 90.85 ↑10.36 | 7.51 | 87.80 ↑4.88 | 6.05 |
| | DoT-bank | 92.68 ↑12.19 | 9.14 | 90.24 ↑7.32 | 8.17 |
| | ParamAgent | 92.07 ↑11.58 | 11.90 | 93.90 ↑10.98 | 8.93 |
| | ParamAgent-plus | **95.03** ↑14.54 | 19.47 | **95.12** ↑12.20 | 16.81 |

Figure 6. For each problem, GPT-4o-mini is further asked to produce potential mistakes along with buggy implementations. This yields a dataset of reflective signals and corresponding erroneous code examples. We then finetune LLaMA-3.1-8B with LoRA on this dataset to obtain the programming-specific parametric module $M_r$.

**Math** For mathematical reasoning, we leverage the MATH training set (Hendrycks et al., 2021b). From each subject area, we randomly sample 800 problems and adopt the same pipeline as in programming: GPT-4o-mini is prompted to produce reflective feedback and buggy derivations for each sampled problem. The resulting dataset is used to LoRA-finetune LLaMA-3.1-8B to instantiate $M_r$ for math reasoning.

**Multi-hop QA** For multi-hop QA, we randomly sample 10000 instances from the HotpotQA (Yang et al., 2018) and 2WikiMultiHopQA (Ho et al., 2020) training sets respectively. GPT-4o-mini is prompted to output structured semantic units (e.g., entities, relations, constraints, answer types, and sub-questions) for each example. We then apply LoRA finetuning to LLaMA-3.1-8B on this dataset to build the parametric module $M_p$.

Across all domains, during dataset construction we provide one carefully designed demonstration example in the prompt to GPT-4o-mini. This ensures that the generated outputs (reflective feedback, buggy code, or semantic units) adhere to the required format, making the synthetic supervision more reliable.

### B.3. How does **ParamAgent** perform with stronger base LLMs?

We further study the performance of ParamAgent when paired with stronger base models of around 70B parameters. Specifically, we use Llama-3.1-70B and Qwen2.5-72B-Instruct as the underlying LLMs, while keeping the parametric module fixed as Llama-3.1-8B. We evaluate on HumanEval for programming and HotpotQA for multi-hop QA. The results are reported in Table 6 and Table 7 respectively.

**Results.** Across tasks, ParamAgent achieves performance that is on par with, or even surpasses, state-of-the-art baselines. Moreover, ParamAgent-plus consistently outperforms the best baseline methods by a large margin, highlighting the effectiveness of the parametric module. It is worth noting that our parametric module itself is only an 8B model, yet it integrates effectively with base LLMs as large as 70B. This demonstrates the strong potential of our approach when scaled further.

*Table 7.* Performance on HotpotQA dataset. **Bold** denotes the best result, and underline marks the second best. ↑ and ↓ indicate the absolute improvement or decrease relative to the Base method. For clarity, the prompt token usage of the Base method is normalized to 1.

| Dataset | Method | Llama-3.1-70B-Instruct | | Qwen2.5-72B-Instruct | |
|---------|--------|------|------|------|------|
| | | Acc | #Prompt Tokens | Acc | #Prompt Tokens |
| HotpotQA | Base | 70.00 | 1.00 | 73.33 | 1.00 |
| | Model-based CoT | 73.67 ↑3.67 | 1.43 | 74.10 ↑1.05 | 1.44 |
| | Reflexion | 82.33 ↑12.33 | 3.02 | 82.67 ↑9.34 | 2.81 |
| | DoT | 73.67 ↑3.67 | 3.43 | 80.67 ↑7.34 | 4.30 |
| | DoT-bank | 80.00 ↑10.00 | 5.24 | 82.33 ↑9.00 | 7.87 |
| | `ParamAgent` | 84.00 ↑14.00 | 7.70 | 81.00 ↑7.67 | 7.90 |
| | `ParamAgent-plus` | **89.67** ↑19.67 | 13.69 | **84.67** ↑11.34 | 15.43 |

*Table 8.* Token usage and cost on HumanEval and HotpotQA datasets with Llama3.1-8B as backbone LLM. Best and second-best metrics are in **bold** and underline respectively.

| Method | HumanEval | | | | HotpotQA | | | |
|--------|-----------|-----------|-----------|--------|-----------|-----------|-----------|--------|
| | #Prompt Tokens | #Completion Tokens | Total Cost ($) | Pass@1 (%) | #Prompt Tokens | #Completion Tokens | Total Cost ($) | Acc (%) |
| Base | 37,463 | 13,506 | 0.00917 | 59.15 | 164,013 | 1,801 | 0.02985 | 57.67 |
| Model-based Reflection | 342,805 | 82,280 | 0.07652 | 78.05 | 236,548 | 1,212 | 0.04280 | 61.67 |
| Reflexion | 348,068 | 73,538 | 0.07589 | 76.22 | 703,192 | 68,612 | 0.13892 | 71.33 |
| DoT | 653,981 | 169,986 | 0.14831 | 72.56 | 1,164,812 | 106,806 | 0.22889 | 66.67 |
| DoT-bank | 926,047 | 233,016 | 0.20863 | 79.88 | 2,179,148 | 195,283 | 0.42740 | 72.00 |
| `ParamAgent` | 814,627 | 163,257 | 0.17602 | **82.93** | 3,649,598 | 128,010 | 0.67997 | **78.33** |

## B.4. Cost Analysis

Table 8 reports prompt/completion tokens and costs using Llama-3.1-8B. Costs are computed with TogetherAI pricing as of Aug 20, 2025 ($0.18 per million tokens). We can see that Model-based Reflection (CoT) is highly efficient, achieving strong accuracy with far fewer tokens than reflection-heavy methods like DoT-bank. By contrast, `ParamAgent` delivers the best results on both HumanEval and HotpotQA, at higher but still moderate cost, this highlights the advantages of incorporating various forms of memory modules.

## B.5. A Case Study

We present a case study from the MBPP dataset, where both Reflexion and DoT fail to generate the correct implementation, while `ParamAgent` succeeds. To better understand this difference, we analyze the reflective history of all three methods and highlight the gists, as illustrated in Figure 8.

From the analysis, we observe that Reflexion and DoT often produce unhelpful sometimes even misleading reflections, which push the agent further away from the correct solution. In contrast, `ParamAgent` generates fewer such misleading reflections. We hypothesize that this advantage arises from the parametric knowledge encoded in $M_r$, which helps `ParamAgent` avoid unhelpful or error-prone reflective signals.

## B.6. Prompt Templates

We provide prompt templates used in `ParamAgent` across different domains. The 1-shot reflective example for programming tasks can be found in Figure 9, and the corresponding math reasoning template in Figure 10. For multi-hop QA, the semantic decomposition 1-shot example is shown in Figure 11.

Instruction templates for generating actions for the programming is shown in Figure 12, the math reasoning instruction in Figure 13, and the multi-hop QA instruction in Figure 14.

```
1  CATEGORIES = [
2      # Core Text & Parsing
3      "String Manipulation",
4      "R e g u l a r Expression Parsing",
5      "Natural-Language Tokenisation",
6      "CSV / JSON Parsing",
7      "URL / URI Parsing",
8      "Text Justification / Word-Wrapping",
9      # Lists, Arrays, SEQ
10     "Array / List Algorithms",
11     "Two-Pointer / Sliding-Window",
12     "Sorting & Searching",
13     "Statistical Summary of Sequences",
14     # Maths & Numbers
15     "Elementary Arithmetic / Algebra",
16     "Number Theory & Divisibility",
17     "Bitwise Operations",
18     "Combinatorics & Counting",
19     "Probability / Statistics",
20     # Data-Structures
21     "Hash / Set / Dict Operations",
22     "Stack / Queue Simulation",
23     "Linked-List Manipulation",
24     "Matrix Operations",
25     "Heap / Priority Queue Operations",
26     "Trie / Prefix-Tree",
27     # Graphs & Trees
28     "Graph / Tree Traversal",
29     "Binary Search Trees",
30     "Dynamic Programming",
31     "Recursion / Backtracking",
32     "Union-Find / Disjoint Set",
33     # Geometry / Coordinates
34     "Geometry & Coordinate Computation",
35     # Dates / Times / Calendars
36     "Date & Time Calculations",
37     # Miscellaneous Practical
38     "File & Path Utilities",
39     "Data-Type Conversion & Formatting",
40     "Cipher / Encoding",
41     "Simulation / Game Logic",
42     "Misc Small-Scale Algorithms"
43 ]
```

*Figure 6.* Schema of categories for synthesizing programming tasks used in our parametric module construction.

```
1  system_content = (
2      "You are an expert Python engineer crafting coding problems.\n"
3      "Follow this EXACT format:\n\n<template_example>\n\n"
4      "- Randomly pick ONE category from the list above.\n"
5      "- Output EXACTLY two lines:\n"
6      "    func_sign: <signature with colon>\n"
7      "    docstring: '<single-quoted string with \\n escapes>'\n"
8      "- Do NOT wrap in JSON or triple quotes.\n"
9      "- Avoid any collisions with past tasks.\n\n"
10 )
```

*Figure 7.* Prompt for synthesizing programming tasks

---

**Reflexion.**
**Problem:** "The implementation failed the test case where it tries to return a single element that is not a list. This issue arises because the ^ operator in the set difference operation returns a set, and then we explicitly convert it to a list..."
**Problem:** "The implementation failed the test where it is expected to find a difference in the lists because the list to check for difference is a subset of the original list... This would return the union of the two lists, not the actual difference."
**Problem:** "The implementation failed the test where the difference should be a single element because it uses the symmetric difference operation which is not what's intended here... When converting the set back to a list using the list function, the order of elements is lost."
**Problem:** "When converting the set back to a list using the list function, the order of elements is lost."

---

**DoT.**
**Problem:** "The implementation failed the test due to a mismatch in the type of the function return value and the expected output."
**Problem:** "The function implementation does not handle the case where the input lists are not of the same length or contain duplicate elements."
**Problem:** "The function implementation does not correctly calculate the symmetric difference between two lists."
**Problem:** "The implementation failed due to a mismatch in the type of the function return value and the expected output, but the proposed fix in the previous reflection is incomplete."
**Problem:** "The function implementation does not correctly handle the case where the input lists are not of the same length or contain duplicate elements..."
**Problem:** "The function implementation does not correctly remove duplicates from the input lists, leading to incorrect results."

---

**ParamAgent.**
**Problem:** "The function implementation does not handle the case of empty lists correctly." **Fix:** "Add a check at the beginning of the function to return an empty list if either of the input lists is empty..."
**Problem:** "The function implementation does not preserve the original order of elements in the lists." **Fix:** "Instead of using set operations..., use a list comprehension to create the difference list..."
**Problem:** "The function implementation does not handle non-integer values in the lists correctly." **Fix:** "Add a check at the beginning... ensure that both input lists are lists of integers..."
**Problem:** "The function implementation does not handle non-list types correctly." **Fix:** "Add a check... ensure that both input arguments are lists..."
**Problem:** "The function implementation does not handle duplicates correctly." **Fix:** "Use a list comprehension... and preserve the original order..."
**Problem:** "The function implementation does not handle the case where the lists are identical." **Fix:** "Return an empty list if the two input lists are identical."

---

*Figure 8.* Selected reflections on the `mbpp_769_Diff` problem. **Red** = critical errors that directly invalidate the implementation; **Orange** = conceptual pitfalls or misleading reasoning patterns; **Green** = key insights leading to a working solution; **Gray** = background noise (irrelevant or stylistic).

```
1  [Function Signature]:
2  def has_close_elements(numbers: List[float], threshold: float) -> bool:
3      """Check if any two numbers in the list are closer than the threshold."""
4
5  [Potential mistakes]:
6  1. **Empty or Single-Element Lists** must return `False`, not `True`.
7  2. **Duplicate Values** must be compared (difference 0), so never drop duplicates.
8  3. Always use **absolute difference** (`abs(a - b)`), not raw subtraction.
9  4. Use the correct **strictness** (`< threshold`, not `<=`).
10 5. Ensure you d o n t **exit too early**  check  all distinct pairs.
11
12 [Flawed Implementations Illustrating Each Pitfall]:
13
14 def has_close_elements_v1(numbers: List[float], threshold: float) -> bool:
15     # BUG: returns True for empty or single-element lists
16     if len(numbers) < 2:
17         return True
18     for i in range(len(numbers)-1):
19         for j in range(i+1, len(numbers)):
20             if abs(numbers[i] - numbers[j]) < threshold:
21                 return True
22     return False
23
24 def has_close_elements_v2(numbers: List[float], threshold: float) -> bool:
25     # BUG: removes duplicates, so identical values never compared
26     numbers = sorted(set(numbers))
27     for i in range(len(numbers)-1):
28         if abs(numbers[i+1] - numbers[i]) < threshold:
29             return True
30     return False
31
32 def has_close_elements_v3(numbers: List[float], threshold: float) -> bool:
33     # BUG: uses raw subtraction instead of abs()
34     for i in range(len(numbers)-1):
35         for j in range(i+1, len(numbers)):
36             if (numbers[i] - numbers[j]) < threshold:
37                 return True
38     return False
39
40 def has_close_elements_v4(numbers: List[float], threshold: float) -> bool:
41     # BUG: uses <= instead of <, misclassifies exactly-threshold pairs
42     for i in range(len(numbers)-1):
43         for j in range(i+1, len(numbers)):
44             if abs(numbers[i] - numbers[j]) <= threshold:
45                 return True
46     return False
47
48 def has_close_elements_v5(numbers: List[float], threshold: float) -> bool:
49     # BUG: breaks out of outer loop too soon
50     ... (omit due to limited page)
51
52 END OF EXAMPLE
```

*Figure 9.* 1-shot example for reflective dataset construction for programming task.

**Question.** Circle $O$ is located on the coordinate plane with center at $(2, 3)$. One endpoint of a diameter is at $(-1, -1)$. What are the coordinates of the other endpoint of this diameter? Express your answer as an ordered pair.

**Pitfalls & Potential Mistakes**

1. **Confusing the center with an endpoint.** Assuming the center is an endpoint leads to an incorrect reflection point.

2. **Incorrect use of the midpoint formula.** Forgetting that the center is the midpoint of the diameter, or solving $(x + x_2)/2 = \text{center}_x$ incorrectly.

3. **Using the wrong coordinates for the midpoint.** Plugging endpoint coordinates in place of the center (or vice versa) yields the wrong unknowns.

4. **Arithmetic errors.** Sign or algebra mistakes when solving, e.g. $2 = (-1 + x)/2 \Rightarrow x = 3$ (incorrect) instead of $x = 5$.

5. **Switching $x$ and $y$.** Mixing $x$- and $y$-midpoint formulas, or using $x$ values to solve for $y$.

6. **Incorrect interpretation of the diameter.** Thinking the diameter extends in the same direction from the center; doubling the vector or reflecting in the wrong direction.

*Figure 10.* 1-shot example for reflective dataset construction in math reasoning.

---

**Example 1**
**Question.** Anatoly Maltsev and Valentin Turchin were both from Russia, which of the two is known for his work as a mathematician?

---

**Question Parsing and Intent Extraction**
**Key Components**

- **Entity A:** Anatoly Maltsev — mathematician/logician; contributions in mathematical logic and abstract algebra.

- **Entity B:** Valentin Turchin — computer scientist/philosopher; work in cybernetics and philosophy of science.

- **Implied Relationship:** Comparative inquiry: which individual is more closely associated with mathematics.

- **Answer Type Expected:** Person name (e.g., "Anatoly Maltsev").

- **Reasoning Type:** Comparative factual reasoning.

- **Required Background:** Biographical profiles or retrieved professional records.

---

**Inference Trace**

1. Retrieve factual data about Maltsev's and Turchin's primary academic domains.

2. Classify Maltsev as a mathematician (core contributions to mathematical logic).

3. Classify Turchin as mainly in cybernetics and philosophy.

4. Eliminate Turchin as the primary mathematician.

5. Conclude: **Anatoly Maltsev**.

---

**Disambiguation Note**
Nationality (Russia) does not help differentiate them.

*Figure 11.* 1-shot example used in `ParamAgent` for semantic decomposition dataset construction in multi-hop QA.

You are an AI Python assistant. You will be given some potential pitfalls and several flawed implementations for the coding challenge, as well as your previous implementation of a function, a series of unit-test results, and your self-reflection on your previous implementation. Try to avoid the errors from your previous implementation and the listed pitfalls.

**Instruction:** ALWAYS WRITE your full implementation (restate the function signature).

*Figure 12.* Instruction prompt used by `ParamAgent` to generate next-round solutions for programming tasks.

**You are revising your previous answer to a mathematics problem.**
You will receive:
(1) the original question,
(2) potential mistakes and pitfalls,
(3) your last answer, (4) feedback (Right or Wrong) explaining why that answer was unsatisfactory, and (5) your brief self-reflection on the mistake.

**Respond with:**
1. **Reasoning**: updated step-by-step thoughts.

2. **Answer**: the corrected final result.

**Formatting:** The final answer should be simplified to its simplest form, e.g., $25$, $25_{16}$, $\frac{1}{36}$, etc.

*Figure 13.* Instruction prompt used by `ParamAgent` to generate next-round solutions for math reasoning.

You are revising your previous answer to a multi-hop QA question.
You will receive:
(1) the original question,
(2) some key points, the underlying intent, and possible inference patterns that facilitate answering this question,
(3) your last answer,
(4) supporting context,
(5) feedback (Right or Wrong) explaining why that answer was unsatisfactory,
(6) your brief self-reflection on the mistake.

**Instruction:** Based on the inputs, produce a new single-phrase answer that resolves the error and fully answers the question. Output only the answer — no commentary, no code.

*Figure 14.* The prompt of `ParamAgent` to generate next-round answers for multi-hop QA tasks.

