# OpenReview forum: "ParamMem: Augmenting Language Agents with Parametric Reflective Memory"
_ICML.cc/2026/Conference — ICML 2026 regular_

### Official Review · Reviewer_3szr · 2026-03-02

**Soundness:** 4
**Presentation:** 3
**Significance:** 4
**Originality:** 3
**Overall Recommendation:** 5
**Confidence:** 5

**Summary:**

The paper “ParamMem: Augmenting Language Agents with Parametric Reflective Memory”
addresses the limited diversity in self-reflection observed in language agents. Prior approaches
rely on cross-sample memory to store reasoning trajectories from previously solved tasks,
which improves agentic reasoning but lacks scalability. Through empirical analysis, the authors
demonstrate a strong positive correlation between reflective diversity and task success. To
address this, they introduce ParamMem, a parametric memory module that encodes cross-
sample reflection patterns into model parameters, enabling diverse reflection generation via
temperature-controlled sampling. Building on this, the paper proposes ParamAgent, a
reflection-based agent framework that integrates parametric, episodic, and cross-sample
memory to enhance reasoning performance.

**Compliance With Llm Reviewing Policy:**

Affirmed.

**Final Justification:**

I have carefully reviewed the changes and the authors’ responses to the reviewers’ comments. I am satisfied that the suggested changes have been adequately addressed and improve the clarity and presentation of the paper. Based on these revisions, I am comfortable maintaining my original rating.

**Key Questions For Authors:**

- The introduction is well written; however, it would benefit from explicitly outlining the
structure and flow of the paper. Can the authors add a brief paragraph at the end of the
introduction describing the organization of the manuscript?
-In the abstract, the problem statement and proposed solution are not clearly connected.
Can the authors explicitly state how ParamMem addresses the challenge of limited
reflective diversity in language agents?
- The manuscript provides limited details about the underlying LLM models used in the
experiments. Can the authors clarify the model architecture, size, and training or fine-
tuning settings to improve reproducibility?
- In Table 4: Sample Efficiency of ParamMem, the results are promising. However, can the
authors include comparisons with more recent or competitive state-of-the-art methods to
better contextualize the performance gains?
- The paper claims that ParamMem’s diversity-driven objective generalizes better across
distributions. Can the authors discuss potential limitations or failure cases of this
objective, especially in tasks where reflective diversity may not directly correlate with
performance?
- How does the proposed parametric memory differ in practice from non-parametric cross-
sample memory in terms of adaptability and scalability?
- What is the computational and memory overhead introduced by ParamMem compared to
baseline reflection-based agents?
- Can the authors comment on the applicability of ParamMem in real-world, long-horizon
agentic tasks, beyond the benchmark settings evaluated in this work?

**Limitations:**

-The proposed diversity-driven objective assumes that increased reflective diversity
consistently leads to better generalization, which may not hold for all tasks or domains.
-Encoding reflection patterns into model parameters may reduce flexibility when
adapting to rapidly changing or highly domain-specific tasks.

**Strengths And Weaknesses:**

Strengths:
• The introduction is well written and clearly motivates the problem of limited reflective
diversity in language agents. The overall presentation is coherent and easy to follow.
• The paper proposes a novel parametric reflective memory module (ParamMem) that
effectively encodes cross-sample reflection patterns into model parameters, which is a
meaningful contribution beyond non-parametric memory approaches.

Weaknesses:
• The manuscript provides limited discussion of the underlying LLM models, including
their size, architecture, and training configurations, which affects reproducibility.
• Although the experimental results are strong, comparisons with more recent state-of-the-
art methods are limited, particularly in the sample efficiency analysis.

---

> ### Author Rebuttal · Authors · 2026-03-31
>
> We thank Reviewer 3szr for the positive evaluation and the constructive questions. We address each question below.
>
> > **Q1: Paper organization.**
>
> Thank you for the suggestion. We have added an organizational paragraph at the end of the introduction in the revised manuscript, outlining the structure from preliminaries through experiments to conclusions.
>
>
> > **Q2: Connection between problem statement and proposed solution in abstract.**
>
> We have revised the abstract to make this connection more explicit in the draft as following:
>
> "ParamMem addresses this by internalizing reflection patterns across different samples into model parameters, enabling diverse reflection generation through temperature-controlled sampling at inference time."
>
> > **Q3: Model details and training settings for reproducibility.**
>
> We use Llama-3.1-8B-Instruct as the base model with bf16 precision. We apply LoRA adapters to the attention projection layers with rank r = 64, alpha = 128, and dropout = 0.05. Training uses a learning rate of 2e-5 with cosine scheduler and 3% warmup ratio, 3 epochs, and an effective batch size of 32. For inference, we use temperature = 0.7, top-p = 0.9, and max tokens = 2048. We have added these details in the appendix of the revised draft. We will also open-source the training scripts for reproducibility.
>
> > **Q4: Comparisons with more competitive baselines in sample efficiency analysis.**
>
> To address the reviewer's concern, we have extended the sample efficiency comparison (Table 4) to include two Retroformer variants: Retroformer (using only the trained module to generate reflections as in the original paper) and Retroformer+ (using both agent self-reflections and the trained module's reflections, which is a fairer comparison to our approach). The results with 500 training samples are illustrated in **[this table](https://anonymous.4open.science/r/chs1-C767/sample_efficiency.pdf)**, due to character limit.
> The results show that under limited sample size, learning accurate reflections significantly underperforms learning diverse reflections.
>
> > **Q5: Limitations and failure cases of diversity-driven objective.**
>
> We acknowledge two limitations: (1) Token overhead due to the additional diveristy of reflection. (2) Task nature: For some specific tasks such as math, and when training and test distributions are well-aligned (e.g., MATH, where training and test set cover the same 7 subjects equally), accuracy-focused approaches like Retroformer occasionally outperforms diversity-driven ones. However, when this alignment does not hold, diversity-driven approach is more advantageous, as shown across most tasks in our evaluation.
>
> > **Q6: adaptability and scalability comparison.**
>
> **Adaptability**. Non-parametric approaches such as DoT-Bank rely on a retrieval-based memory bank, which limits adaptability to new tasks or domains not well represented in the stored trajectories. In contrast, ParamMem internalizes reflection patterns into model parameters. It requires no labeled data or stronger external model. The gains from combining both memory types in ParamAgent-plus confirm they provide complementary information.
>
> **Scalability.** We observe strong scalability along two axes for parametric reflective memory. For model scale, our experiments confirm effectiveness from 1.5B to 80B parameters. For data scale, only 500 samples suffice for strong performance. This is a notable advantage over cross-sample memory, which depends on retrieving similar trajectories and is thus more sensitive to data quality and coverage. The parametric approach is less reliant on data volume, as the base model's pretraining already encodes broad knowledge that fine-tuning efficiently activates.
>
> > **Q7: Computational and memory overhead**
>
> **Training.** Since ParamMem is trained via LoRA with a small amount of samples, it is computational friendly. For  Llama-3.1-8B, we observe a total training time of approximately 4-5 GPU-hours for 3 epochs.
>
> **Inference.** During inference, each iteration adds one additional forward pass to generate a model-based reflection. While this introduces extra FLOPs per iteration, the added cost is equivalent to generating one more reflection with the base LLM, a modest overhead that is well justified by the consistent performance gains observed across all datasets and domains.
>
> > **Q8: Applicability to real-world, long-horizon agentic tasks.**
>
> Thank you for this forward-looking question. By internalizing diverse reflective patterns into parameters, ParamMem could help agents in long-horizon tasks explore a **wider hypothesis space** at each decision point rather than converging prematurely. This could be particularly valuable in open-ended settings such as software engineering and multi-step planning.
>
> ---
>
> We would be happy to provide further clarification if needed.

---

> > ### Author Rebuttal · Reviewer_3szr · 2026-04-03
> >
> > Thank you for addressing all the concerns and adding detailed responses. I would like to maintain my original score.

---

> > > ### Author Response · Authors · 2026-04-05
> > >
> > > Dear reviewer, thank you again for your positive feedback and for recognizing our contributions. We are very glad to know that our responses have addressed your concerns. We also thank you for your valuable suggestions, which will help us further enhance the paper.

---

### Official Review · Reviewer_53aD · 2026-03-09

**Soundness:** 2
**Presentation:** 2
**Significance:** 2
**Originality:** 2
**Overall Recommendation:** 3
**Confidence:** 4

**Summary:**

This work addresses the problem of repetitive and low-diversity self-reflections in LLM agents, which limit iterative reasoning performance.
The authors first establish a strong positive correlation between reflection diversity and task performance.
Based on this observation, they propose ParamMem, a lightweight parametric memory module that is fine-tuned on a reflection dataset to internalize reflection patterns.
At inference time, ParamMem generates diverse reflections through temperature-controlled sampling, complementing existing episodic and cross-sample memory.
Building on this, they propose ParamAgent and ParamAgent-plus. Experiments across code generation, math reasoning, and multi-hop QA with three backbone LLMs show consistent improvements over baselines. Additional analyses demonstrate sample efficiency, self-improvement capability, and weak-to-strong transfer.

**Compliance With Llm Reviewing Policy:**

Affirmed.

**Final Justification:**

I appreciate the authors' detailed rebuttal. However, the massive discrepancy in token usage remains an unfair comparison, making it impossible to discern whether the performance gains stem from the ParamMem architecture or simply a significantly higher computational budget. Furthermore, the argument that diversity is primarily achieved by including outcome-agnostic (incorrect) data is not sufficiently persuasive to demonstrate a fundamental architectural advantage over Retroformer's near-on-policy exploration. Without experiments that normalize the token budget or isolate the data filtering strategy, the core claims regarding ParamMem’s efficiency and the necessity of its parametric design remain unconvincing. Consequently, I maintain my original assessment as the soundness of the comparative analysis has not been adequately resolved.

**Key Questions For Authors:**

1. Have you compared ParamMem against a baseline that uses the same compute budget for rejection sampling followed by SFT on successful trajectories from Reflexion? If such a simpler approach achieves comparable results, the justification for the proposed parametric memory module would need to be reconsidered.
2. Does the proposed method explicitly optimize for reflective diversity, or does diversity emerge only as a side effect of parametric generation and temperature sampling? If the latter, have you considered incorporating a diversity-aware objective, and would this further improve performance?
3. Do the improvements hold on more challenging benchmarks such as AIME, SWE-bench, or MuSiQue? If the gains diminish on harder problems, this would significantly affect the assessment of ParamMem's practical impact.

**Limitations:**

The paper already includes limitations in Section 6.

**Strengths And Weaknesses:**

### Strengths

- The motivation is compelling: the observation that reflective diversity is strongly correlated with task performance (Figure 1) provides a clear and well-grounded rationale for the proposed approach. The idea of using a parametric memory module to generate diverse reflection hints, rather than relying on prompt variations or retrieval alone, is a sensible and interesting direction.
- Observations 4, 5, and 6 (self-improvement without stronger external models, iterative self-teaching, and weak-to-strong transfer) are interesting and meaningful contributions that go beyond the main performance results, demonstrating the practical versatility of ParamMem.

### Weaknesses

- The presentation quality needs significant improvement. Figure 2 attempts to illustrate three different frameworks (Reflexion, DoT-bank, ParamAgent-plus) in a single figure using color coding (Pale Mint, Lavender Gray, Soft Sand), which makes it very difficult to parse. A side-by-side comparison of the existing approach vs. the proposed approach would be far more effective. Additionally, the notation is unnecessarily complex: the paper introduces an excessive number of symbols ($p_\theta$, $p_\psi$, $M_g$, $r_k^g$, $\tau_{1:j}$, $f_\phi$, $B$, $D$, $P$, etc.), and Equations 1-6 each describe essentially the same generation process with slightly different conditioning inputs, which obscures rather than clarifies the core idea.
- The evaluation is limited to relatively easy and standard benchmarks. To convincingly demonstrate that ParamMem's reflections generalize, experiments on more challenging and recently adopted benchmarks would be necessary: for math, AIME or OlymMATH [1]; for code, SWE-bench or Terminal-Bench [2]; for QA, MuSiQue [3] or similar datasets requiring 3+ hops that are not seen during training. Without this, it is unclear whether the improvements hold on problems where reflection quality matters most.
- The comparison is not entirely fair in terms of compute budget. Most baselines (Reflexion, DoT, DoT-bank) are prompting-based and require no additional training, while ParamMem requires LoRA fine-tuning on an auxiliary dataset. A natural baseline to consider would be using the same compute budget for rejection sampling to collect successful trajectories with Reflexion and then supervised fine-tuning the target model directly. The absence of such a comparison makes it difficult to isolate the value of the proposed parametric memory approach from the benefit of simply having additional training.
- The paper's central motivation is that diversity is key to improving reflection, yet the proposed method does not include any mechanism that explicitly optimizes for diversity. ParamMem is simply a LoRA module trained with standard cross-entropy on gold reflections, and diversity arises only incidentally through temperature sampling. This raises a concern about novelty relative to Retroformer, which also trains a parametric reflection module. If neither method explicitly targets diversity, the primary difference reduces to the optimization objective (policy gradient vs. cross-entropy), which is a limited contribution.

#### References

[1] Sun et al., Challenging the Boundaries of Reasoning: An Olympiad-Level Math Benchmark for Large Language Models

[2] Merrill et al., Terminal-Bench: Benchmarking Agents on Hard, Realistic Tasks in Command Line Interfaces

[3] Trivedi et al., MuSiQue: Multihop Questions via Single-hop Question Composition

---

> ### Author Rebuttal · Authors · 2026-03-31
>
> We thank Reviewer 53aD for the detailed feedback. We address each concern below.
>
> ---
>
> > **W1: Presentation quality and notation.**
>
> We have revised the manuscript to address all presentation concerns:
>
> - (1) Figure 2 now includes explicit text labels within each region, removing reliance on color coding alone;
>
> - (2) we unified $f_\phi$ and $p_\phi$ into a single symbol $p_\phi$, and the remaining distinct symbols are necessary to precisely describe the conditioning for our three memory types;
>
> - (3) we added forward references between Equations 1–6 to make their progressive structure explicit (Eqs 1–2: prior formulations, Eqs 3–4: our proposed conditioning, Eqs 5–6: full formulation).
>
> > **W2: Evaluation on more challenging benchmarks.**
>
> **Code generation.** We note that our submission already includes **LiveCodeBench (Table 3 in the draft)**, a recently adopted and more challenging benchmark that features competition-level coding problems with continuous updates to avoid data contamination. ParamAgent achieves 64.67% and ParamAgent-plus achieves 68.00%, outperforming all baselines including DoT-Bank and Reflexion, even using a weaker parametric reflection model (Qwen3-Next-30B-A3B-Instruct).
>
> **Math and QA.** To further address this concern, we have conducted additional experiments on two harder benchmarks suggested by the reviewer: **OlympMATH** (130 olympiad-level math problems) and **MuSiQue** (multi-hop QA requiring 2–4 hops, 75 samples per hop level). We use Qwen3-30B-A3B as the base model and fine-tune ParamMem on its own self-generated reflection data. The experimental results can be found below (per-hop accuracy results for MuSiQue are shown in **[this table](https://anonymous.4open.science/r/chs1-C767/musique_table.pdf)**):
>
> Table: Performance comparison on OlympMATH and MuSiQue.
> | Method | OlympMATH | MuSiQue |
> |---|:---:|:---:|
> | Base | 17.69% | 33.8% |
> | Reflexion | 27.69% | 44.0% |
> | DoT | 29.23% | 35.6% |
> | DoT-Bank | 32.31% | 39.6% |
> | Retroformer (w/ correct reflections) | 28.46% | -- |
> | ParamAgent | 33.85% | 35.6% |
> | ParamAgent-plus | **36.15%** | **47.1%** |
>
> Even on these more challenging datasets, ParamMem remains effective: ParamAgent-plus consistently outperforms all baseline methods, achieving +3.84% over DoT-Bank on OlympMATH and +3.1% on MuSiQue overall. Notably, on MuSiQue 4-hop questions (the hardest setting, not seen during training), ParamAgent-plus achieves 45.3%, significantly outperforming all baselines.
>
>
> > **W3: Compute budget fairness.**
>
> We clarify that our main experiments already include a fair compute-matched comparison through **Retroformer**. In Table 1 of the draft, both ParamMem and Retroformer use the same amount of GPT-4o-mini generated data and fine-tune models with the same parameter count. Retroformer collects successful trajectories paired with environment feedback and optimizes via PPO. Under this controlled setting, our method outperforms Retroformer in the majority of cases across all three domains.
>
> Furthermore, we have directly tested the exact baseline the reviewer suggests: on OlympMATH, we use rejection sampling to collect successful reflection trajectories from Reflexion, then supervised fine-tune the target model on these correct reflections. As shown in the table above  this approach (Retroformer w/ correct reflections) achieves only 28.46%, underperforming both ParamAgent (33.85%) and ParamAgent-plus (36.15%). This result implies that optimizing for reflection correctness is less effective than optimizing for diversity, supporting our claim.
>
>
> > **W4: Motivation clarification relative to Retroformer and diversity mechanism.**
>
> We respectfully clarify that the difference between ParamMem and Retroformer will not reduce to merely the optimization objective. The two methods differ fundamentally in both the data collection process, annotation method and the optimization goal.
>
> **Retroformer** requires supervised signals: it first collects correct and incorrect reflections paired with environment feedback, then trains a reflection module via PPO to generate reflections that maximize task success. Its optimization goal is **accuracy**, i.e., generating reflections that are as correct as possible given the problem and context.
>
> **ParamMem**, in contrast, requires no supervised signal during data collection. We simply prompt an LLM to generate reflections across different samples and use these as training data for standard SFT. The key insight is that by fine-tuning on reflections from diverse samples, the model implicitly learns to capture cross-sample reflection patterns. Our optimization goal is **diversity**, not correctness. Figure 4 in the draft empirically supports this: the SFT-finetuned module generates more diverse reflections than the frozen base model.
>
>
> ---
>
> We would be happy to provide further clarification if needed.

---

> > ### Author Rebuttal · Reviewer_53aD · 2026-04-03
> >
> > I appreciate the authors’ rebuttal and their efforts to revise the paper to address my concerns regarding the presentation. However, I have unresolved concerns regarding the experimental setup and core claims that persist even after reading the rebuttal.
> >
> > **1. Contradiction regarding Retroformer and Diversity (W1 vs. W4)**
> >
> > The authors stated that ParamMem and Retroformer were trained on the identical GPT-4o-mini dataset (W3), yet later described Retroformer as utilizing environment feedback and PPO (W4). I interpret this to mean that PPO does not rely on on-policy generated reflections, but rather utilizes the same off-policy static reflections generated by GPT-4o-mini.
> >
> > If the exact same static dataset was used for both, the only variable is the training objective (Cross-Entropy vs. Policy Gradient). There is no inherent evidence that SFT yields more diverse outputs than Policy Gradient. To support your claim, please provide a direct diversity comparison (similar to Figure 4) between ParamMem and Retroformer.
> >
> > **2. Unfair Comparison in Token Usage (Table 1)**
> >
> > There is a massive discrepancy in prompt token counts. For example, in the MBPP (Llama-3.1-8B) task, Retroformer uses 8.64 tokens while ParamAgent uses 86.39 tokens in the most significant case. This order-of-magnitude difference suggests that ParamAgent receives significantly richer context or operates within a different agent loop. Please clarify whether the prompt structures differ and justify how this constitutes a fair comparison for isolating the effects of parametric diversity. I cannot find any detailed explanation regarding this in the paper.
> >
> > **3. Inconsistent Backbones in New Experiments**
> >
> > While the new OlympMATH and MuSiQue results are appreciated, Qwen3-30B is only utilized. The main paper evaluates much smaller models (Llama-3.1-8B, Mistral-7B, Qwen2-1.5B). To demonstrate that the method itself drives performance gains on harder tasks rather than the results being carried by the strength of the 30B backbone, please provide results for these new benchmarks using at least one of the original backbones (e.g., Llama-3.1-8B).
> >
> > I will maintain my current score until above concerns are clearly resolved.

---

> > > ### Author Response · Authors · 2026-04-05
> > >
> > > Dear reviewer, thank you for your careful review and thoughtful questions. We address your concerns below.
> > >
> > >  > ### **Q1: Diversity Comparison**
> > >
> > > We wish to further clarify the distinction between ParamMem and Retroformer as follows:
> > >
> > > **1. Data.** ParamMem trains on outcome-agnostic reflections $\mathcal{D} = \{(x\_i, r\_i^g)\}$ with **no correctness filtering**. Retroformer uses successful reflections with **correctness filtering**.
> > >
> > > **2. Training.** ParamMem fits $p\_\psi(r \mid x)$ conditioned on only the input, learning general cross-sample patterns via SFT. Retroformer fits $p\_\Theta(r \mid x, c)$ where $c$ includes environment feedback and reflection history. We clarify that the dataset is also different due to this setup: For ParamMem, the dataset is indeed static, however for Retroformer, the dataset is generated given the context $c$ at each iteration **dynamically**. For W3, we respectively clarify we did not mean the two methods share an identical dataset, we meant both use GPT-4o-mini as the annotator, thus our approach does not have the data advantage over retroformer, the resulting datasets are still different due to the distinct conditioning.
> > >
> > > **3. On-policy learning.** We clarify that Retroformer is **near-on-policy**: We warm-start the Llama3.1-8b model with 2 epochs of SFT on GPT-4o-mini generated reflection, then run standard PPO,  therefore it is not purely off-policy.
> > >
> > > **4. Inference.** Following the original paper, Retroformer utilizes model-based reflections, without self-reflection from the agent itself; In contrast, ParamAgent also preserves the agent's own reflection.
> > >
> > > **Direct diversity evidence.** We provide **[`the figure link`](https://anonymous.4open.science/r/chs1-C767/final_plots_with_retroformer.pdf)** to include Retroformer to address your concern. As shown, Retroformer is even lower than the frozen base model, indicating that the reflction model converges to a narrow "correct" reflection mode for each input. In contrast, ParamMem substantially improves diversity, owing to the differences in data annotation and training formulation described above.
> > >
> > >  > ###  **Q2: Token Usage Discrepancy**
> > >
> > > We clarify that the comparison is fair at the **training** and **inference configuration** level: both methods use the same max iterations ($T\_{max} = 5$). The token gap arises from two factors inherent to the method designs:
> > >
> > > **1. Reflection source.** As discussed in Q1 (Inference), ParamAgent uses **both** self-reflections and model-based reflections (Algorithm 1, Line 7), while Retroformer uses **only** its module's reflections following the original paper. This additive design naturally increases prompt length.
> > >
> > > **2. Iteration depth.** We have analyzed MBPP coding task, Retroformer solves **87.1% of its successful tasks (samples) on the first attempt**, generating reflection only once. For ParamAgent, **57.5% of all tasks are solved through multi-round iterations**. The prompt context grows across iterations as reflection and implementation history accumulates. However, this is not a controllable variable but rather **a consequence of each method's design**. This also highlights the fundamental difference between the two approaches: Retroformer learns more *accurate* reflections that resolve some tasks, while ParamAgent leverages more *diverse* reflections to iteratively explore the solution space, trading additional token cost for stronger performance.
> > >
> > > > ### **Q3: Results on Harder Benchmarks with Smaller Backbones**
> > >
> > > We chose Qwen3-30B because OlympMATH is extremely challenging for smaller models, the Llama-3.1-8B base achieves only **1.54%** accuracy, making it difficult to meaningfully evaluate method differences. We therefore used a stronger backbone to validate the approach on harder tasks. To address the reviewer's concern, we provide Llama-3.1-8B results on MuSiQue and OlympMATH, **as shown in [`the table link`](https://anonymous.4open.science/r/chs1-C767/tables_llama8b_results.pdf)** due to the character limit.
> > >
> > > Here Retroformer+ uses both self-reflections from the agent and reflections from the Retroformer module, serving as a fairer baseline to our approach. We do not include DoT-Bank and ParamAgent-plus for OlympMATH because the number of positive (solved) samples is too limited to construct a meaningful cross-sample memory bank. For OlympMATH, all methods show limited absolute performance due to the extreme difficulty of competition-level mathematics for Llama3.1-8B, yet ParamAgent still achieves the highest accuracy, demonstrating a consistent advantage even in this challenging regime. On MuSiQue, ParamAgent-plus achieves the best overall performance with particularly strong gains on harder 4-hop questions (**+18.67%** over Retroformer+), confirming that the method itself drives the improvement.
> > >
> > > ---
> > >
> > > We hope that these further responses clearly clarify your concerns.

---

### Official Review · Reviewer_vaR8 · 2026-03-13

**Soundness:** 3
**Presentation:** 3
**Significance:** 3
**Originality:** 3
**Overall Recommendation:** 4
**Confidence:** 2

**Summary:**

This paper first establishes a strong positive correlation between reflection diversity and task performance, then proposes ParamMem, a parametric reflection memory module that encodes cross-sample reflection patterns into model parameters via LoRA fine-tuning. Building on this, the authors construct two agent frameworks: ParamAgent and ParamAgent-plus. Experiments across code generation, mathematical reasoning, and multi-hop QA show that both ParamAgent and ParamAgent-plus achieve better accuracy and higher reflection diversity compared to baselines such as DoT-bank.

**Compliance With Llm Reviewing Policy:**

Affirmed.

**Final Justification:**

I thank the authors for their detailed response, which has addressed my concerns, and I will maintain my positive score.

**Key Questions For Authors:**

Why is the correlation between reflection diversity and task success rate much lower on the MBPP dataset? Does this suggest that the positive correlation between reflection diversity and performance may depend on the task domain?

**Limitations:**

yes

**Strengths And Weaknesses:**

Strengths:

1. Long-term knowledge updating remains an important challenge for large language model systems. Compared with existing prompt-based or retrieval-based approaches, the paper offers an interesting idea of using parameterized external memory through ParamMem.

2. ParamMem is implemented using LoRA-based fine-tuning without modifying the base model. It can be integrated into existing Reflexion-style frameworks and has relatively low training cost with a modular design.

3. The paper evaluates the method on three task domains with different characteristics and compares it with several baselines. It also presents several empirical observations, and the overall experimental setup is reasonably comprehensive.


Weaknesses:

1. The reliability of the reported correlation between reflection diversity and task success rate is unclear. The correlation analysis is based on only three methods, resulting in three data points per dataset, which limits statistical significance. In addition, the correlation on the MBPP dataset (r = 0.22) is much lower than that on the other datasets (0.75–0.97).

2. ParamAgent and ParamAgent-plus combine multiple memory components, but the paper does not provide ablation studies to isolate the individual contribution of each memory component.

---

> ### Author Rebuttal · Authors · 2026-03-31
>
> We sincerely thank Reviewer 1 for the thorough and constructive feedback. We address each concern below.
>
> ---
>
> > **Q1: The correlation on the MBPP dataset (r = 0.22) is much lower than on other datasets.**
>
> We appreciate the reviewer for pointing this out. Upon careful re-examination, we identified a data-processing error in our original Figure 1: the DoT-Bank data point for MBPP mistakenly used the **Base** accuracy (47.61%) instead of the correct DoT-Bank accuracy (64.82%). This erroneous data point is the direct cause of the anomalously low r = 0.22 on MBPP.
>
> After correcting the DoT-Bank data point to (cosine distance = 0.341, accuracy = 0.648), the per-dataset correlation for MBPP with the original three methods increases to r = 0.49. Furthermore, we have extended the analysis by including our proposed method ParamAgent (cosine distance = 0.358, accuracy = 0.670) as a fourth data point across all five datasets. With this correction and addition, the MBPP correlation rises to r = 0.76, and the corrected average Pearson r across all five datasets is now **0.86**, strengthening our original motivation that reflective diversity is positively correlated with task performance. We have updated Figure 1 accordingly (**see [updated figure](https://anonymous.4open.science/r/chs1-C767/reflection_diversity.pdf)**). We apologize for this oversight and we have corrected it in the revised manuscript.
>
> ---
>
> > **Q2: The correlation analysis is based on only three methods (three data points per dataset), limiting statistical significance.**
>
> We acknowledge that the original analysis included only three methods per dataset. This was primarily because the exploration of diversity in self-reflection is still under-explored in the literature. Reflexion, DoT, and DoT-Bank are among the very few existing approaches that explicitly address reflective diversity, making it difficult to include more baselines in this analysis.
>
> To address this concern, we have now extended the correlation analysis by including our proposed method ParamAgent as a fourth data point across all five datasets. With four methods per dataset, the per-dataset correlations remain consistently strong (r values ranging from 0.76 to 0.94), and the average r improves from 0.76 to **0.86**.
>
> Furthermore, we emphasize that this correlation analysis spans **five diverse datasets** across three distinct domains. The consistency of the positive correlation across all five datasets provides stronger cross-domain evidence for the relationship between reflective diversity and performance than would be achieved by analyzing more methods on a single dataset.
>
> ---
>
> > **Q3: The paper does not provide ablation studies to isolate the individual contribution of each memory component.**
>
> We would like to clarify that the existing experimental design already constitutes an implicit ablation study over memory components. Specifically, the baselines and our proposed methods correspond to systematic combinations of three memory types:
>
> | Method | Episodic Memory | Cross-Sample Memory | Parametric Memory |
> |---|:---:|:---:|:---:|
> | Reflexion | Yes | - | - |
> | DoT | Yes | - | - |
> | DoT-Bank | Yes | Yes | - |
> | **ParamAgent** (ours) | Yes | - | Yes |
> | **ParamAgent-plus** (ours) | Yes | Yes | Yes |
>
> This progression isolates the contribution of each component:
>
> - **Episodic memory** is the shared foundation across all methods (Reflexion, DoT serve as the episodic-only baselines).
> - **Cross-sample memory**: Comparing DoT-Bank (episodic + cross-sample) vs. DoT (episodic only) isolates the contribution of cross-sample memory.
> - **Parametric memory**: Comparing ParamAgent (episodic + parametric) vs. Reflexion/DoT (episodic only) isolates the contribution of our proposed ParamMem module.
> - **Full integration**: ParamAgent-plus (episodic + parametric + cross-sample) vs. DoT-Bank (episodic + cross-sample) isolates the added value of parametric memory on top of cross-sample memory.
>
> As shown in Table 1 of our paper, each memory component provides consistent gains: ParamAgent outperforms Reflexion and DoT across all datasets, confirming the standalone effectiveness of parametric memory; ParamAgent-plus further outperforms DoT-Bank, demonstrating the complementary value of combining parametric memory with cross-sample memory. We will clarify this ablation in the experiment section of the revised paper.
>
> ---
>
> We would be happy to provide further clarification if needed.

---

> > ### Author Rebuttal · Reviewer_vaR8 · 2026-04-03
> >
> > I thank the authors for their detailed response, which has addressed my concerns, and I will maintain my positive score.

---

> > > ### Author Response · Authors · 2026-04-05
> > >
> > > Dear reviewer, thank you for your positive feedback and for recognizing our contributions. We are very glad to know that our responses have addressed your concerns. We also sincerely thank you for helping us identify errors in the paper, which gave us the opportunity to correct and improve it.

---

### Decision · Program_Chairs · 2026-04-30

**Decision:**

Accept (regular)

**Comment:**

This paper proposes ParamMem, a parametric reflective memory module for language agents, together with ParamAgent and ParamAgent-plus. The main idea is to encode cross-sample reflection patterns into model parameters so that the agent can generate more diverse reflections at inference time.
The paper has a meaningful and fairly original core idea. Moving from non-parametric reflection memories toward a lightweight parametric memory module is an interesting direction, and the experiments show consistent gains across multiple domains. I also found the additional analyses on sample efficiency, self-improvement, and weak-to-strong transfer useful, since they make the contribution broader than just one benchmark table. The rebuttal improved the paper in important ways. In particular, the authors corrected and strengthened the reflection-diversity analysis, clarified the relation to Retroformer, added harder-benchmark results, and provided more details on model settings and training. These additions make the empirical case stronger.
The main remaining concern is fairness of comparison, especially around token usage and the exact relationship between ParamMem and training-based reflection baselines such as Retroformer. I think this concern is legitimate and should be reflected in the final paper more clearly.